# Transcriptomic Analysis on the Peel of UV-B-Exposed Peach Fruit Reveals an Upregulation of Phenolic- and UVR8-Related Pathways

**DOI:** 10.3390/plants12091818

**Published:** 2023-04-28

**Authors:** Marco Santin, Samuel Simoni, Alberto Vangelisti, Tommaso Giordani, Andrea Cavallini, Alessia Mannucci, Annamaria Ranieri, Antonella Castagna

**Affiliations:** 1Department of Agriculture, Food and Environment, University of Pisa, Via del Borghetto 80, 56124 Pisa, Italy; 2Interdepartmental Research Center Nutrafood ‘‘Nutraceuticals and Food for Health’’, University of Pisa, Via del Borghetto 80, 56124 Pisa, Italy

**Keywords:** flavonoids, metabolomics, *Prunus persica*, secondary metabolism, transcriptomics, UV-B radiation

## Abstract

UV-B treatment deeply influences plant physiology and biochemistry, especially by activating the expression of responsive genes involved in UV-B acclimation through a UV-B-specific perception mechanism. Although the UV-B-related molecular responses have been widely studied in *Arabidopsis*, relatively few research reports deepen the knowledge on the influence of post-harvest UV-B treatment on fruit. In this work, a transcriptomic approach is adopted to investigate the transcriptional modifications occurring in the peel of UV-B-treated peach (*Prunus persica* L., cv Fairtime) fruit after harvest. Our analysis reveals a higher gene regulation after 1 h from the irradiation (88% of the differentially expressed genes—DEGs), compared to 3 h recovery. The overexpression of genes encoding *phenylalanine ammonia-lyase* (*PAL*), *chalcone syntase* (*CHS*), *chalcone isomerase* (*CHI*), and *flavonol synthase* (*FLS*) revealed a strong activation of the phenylpropanoid pathway, resulting in the later increase in the concentration of specific flavonoid classes, e.g., anthocyanins, flavones, dihydroflavonols, and flavanones, 36 h after the treatment. Upregulation of UVR8-related genes (*HY5*, *COP1*, and *RUP*) suggests that UV-B-triggered activation of the UVR8 pathway occurs also in post-harvest peach fruit. In addition, a regulation of genes involved in the cell-wall dismantling process (*PME*) is observed. In conclusion, post-harvest UV-B exposure deeply affects the transcriptome of the peach peel, promoting the activation of genes implicated in the biosynthesis of phenolics, likely via UVR8. Thus, our results might pave the way to a possible use of post-harvest UV-B treatments to enhance the content of health-promoting compounds in peach fruits and extending the knowledge of the UVR8 gene network.

## 1. Introduction

Depending on the wavelength and intensity, light might affect several aspects during plant lifespan. Particularly, UV radiation has been studied over the last decades because of its strong impact on both plant primary and secondary metabolism [1,2]. In the UV range (100–400 nm), UV-B (280–315 nm) represents the most energetic wavelengths reaching the Earth’s surface, since the extremely harmful UV-C is totally shielded by the stratosphere. Therefore, due to its high energy, it may determine severe damage to intracellular components, e.g., nucleic acids and proteins, compromising their physiological functions [3]. However, millions of years of evolution have led plants to adapt toward UV-B conditions because of specific photoreceptors and complex transduction pathways, which in turn determine a fine regulation at gene expression level with repercussions on specific secondary metabolites [4]. In *Arabidopsis*, UV resistance locus 8 (UVR8) has been found to be the photoreceptor responsible for the perception of UV-B and, partially, UV-A radiation [5,6]. The highly energetic UV wavelengths determine the dissociation of the inactive UVR8 dimer into active monomers that activate the downstream signaling cascade by binding the E3 ubiquitin ligase constitutively photomorphogenic 1 (COP1), preventing the elongated hypocotyl 5 (HY5) bZIP transcription factor from being degraded [3,7]. The UVR8-COP1 complex, once translocated within the nucleus, induces a specific chromatin modification, activating the transcription of specific UVR8-responsive genes, which contributes to the acclimation of the plant organism to the UV conditions [8,9]. In addition, the expression of HY5 was also found to be promoted by the UVR8-COP1 complex, amplifying the UVR8-COP1 action. Several genes involved in the phenolic pathway have been found to be strictly regulated by UV-B radiation, such as *chalcone syntase* (*CHS*), *chalcone isomerase* (*CHI*), *flavanone 3-hydroxylase* (*F3H*), *flavonoid 3′-hydroxylase* (*F3′H*), *dihydroflavonol 4-reductase* (*DFR*), and *anthocyanidin synthase* (*ANS*) in different fruit species, e.g., apple, tomato, and peach [10,11,12]. The UV-B signal is then turned off by the action of the UVR8-COP1 complex-activated repressor of UV-B photomorphogenesis 1 (RUP1) and 2 (RUP2), which promote the reconstitution of the UVR8 dimer [13].

Phenolic compounds represent one of the largest classes of plant secondary metabolites, which include more than 8000 members. Phenolics, and particularly flavonoids, play fundamental roles during plant lifespan, contributing to, e.g., the resistance against pests, herbivores, and pathogens, the neutralization of the reactive oxygen species (ROS), the shielding against potentially harmful UV radiations, and in general, the protection against adverse environmental conditions [14,15]. Additionally, once ingested by people through their diet, they exhibit beneficial effects both in prevention and treatment of several diseases, such as cardiovascular diseases, many types of cancers, diabetes, age-related diseases, oxidative stress, and disorders of the immune system [16,17,18,19,20]. The main dietary sources of flavonoids are fruits and vegetables; thus, their consumption is highly encouraged. Additionally, the ever-increasing attention toward food rich in health-promoting compounds, together with the search for eco-friendly technologies, such as UV-B radiation, to increase their content within fruits and vegetables, has gained great popularity [21,22].

Recently, several studies were conducted using whole transcriptome sequencing (RNA-Seq) on the *Prunus* family, exploiting the availability of the complete genome sequence of *Prunus persica* [23], aiming to identify genes involved in biological processes such as cold injuries [24], fruit ripening [25], drought stress [26], and leaf senescence [27].

However, although it is known that UV-B, depending on the dose and duration of the exposure, has the potential to enhance the nutraceutical quality of peach fruit by increasing flavonoid content [12,28], the current studies investigating the molecular response following UV-B exposure on fruit organisms focus on specific flavonoid biosynthetic genes, without investigating the changes in the whole peach transcriptome. With this work, for the first time, the peel transcriptome of UV-B-exposed peach fruit has been investigated by using RNA-Seq, to have an exhaustive overview of the molecular response of the fruit toward the UV-B radiation. In addition, the results from gene networks obtained from transcriptomic analysis have been correlated with metabolomic data, with particular attention to flavonoids.

## 2. Results

### 2.1. cDNA Libraries Aligning on Reference Transcriptome

In total, 339,016,520 sequence reads, each 150 nt in length, from 12 libraries of *P. persica* UV-B-treated (UVB) and control (CTR) fruit peels at two recovery time points (1 and 3 h) were generated. Removal of low-quality reads resulted in 335,883,778 total trimmed reads and the number of reads for each library spanned from 23,126,890 to 32,949,871. The trimmed high-quality reads were aligned on the reference transcriptome of *P. persica*, resulting in a mean of mapping reads of 94%, a reliable percentage of mapped reads to establish gene expression analysis (Table 1).

### 2.2. Differential Expression and Gene Ontology Analysis of Genes of Control and UV-B-Treated Peach Fruit

The 47,089 transcripts included in the *P. persica* genome assembly [23] were evaluated according to their expression. Genes were considered expressed when RPKM > 1 in at least one library. A total of 24,893 expressed genes were selected. Overall, 847 DEGs were found between control and UV-B treated samples at 1 h and 3 h recovery time points. In particular, 1 h after the end of the UV-B treatment, we detected 580 over-expressed genes (OE) and 165 under-expressed genes (UE). At the 3 h recovery time point, 83 OE and 19 UE were found (Figure 1), showing that most of the differential gene expression occurred after 1 h from the UV-B exposure.

Regarding OE genes, 539 genes were uniquely activated after 1 h, and 43 genes were over-expressed solely after 3 h from the end of the UV-B treatment. Interestingly, 40 genes were over-expressed in both the recovery time points (Figure 1). The complete set of DEGs from the peel of control and UV-B-treated fruit is shown in Appendix A.

Under- and over-regulated DEGs, considering 1 h and 3 h recovery, were analyzed by gene ontology (GO) (Appendix A and Appendix A, respectively). Over-expressed DEGs in the two time points showed a very similar GO term distribution. The most abundant terms regarding molecular function were “Catalytic activity” (GO:0003824) and “Binding” (GO:0005488), whereas concerning biological processes, the most frequent GO terms were “Cellular process” (GO:0009987), “Metabolic process” (GO:0008152), “Biological regulation” (GO:0065007), and “Regulation of biological process” (GO:0050789). Furthermore, under-expressed DEGs during the two time points shared few GO terms with over-expressed ones, such as “Catalytic activity” (GO:0003824), “Binding” (GO:0005488), and “Metabolic process” (GO:0008152), but also showed different GO terms belonging to biological processes “Response to stimulus” (GO:0050896) and “Localization” (GO:0051179), cellular component “Membrane” (GO:0016020), and molecular function “Transporter activity” (GO:0005215).

Concerning GO enrichment, 579 over- and 165 under-expressed DEGs after 1 h of UV-B exposure were analyzed (Appendix A). Overall, we detected 14 and 1 enriched GO terms for over- and under-expressed DEGs, respectively. The most abundant enriched GO terms for over-expressed DEGs were “Biological regulation”, “Regulation of cellular process”, and “Aromatic compound biosynthetic process”, whereas only one GO term, “Response to endogenous stimulus”, was identified for under-expressed DEGs at the 1 h recovery time point (Appendix A). No enriched GO terms were found for over- and under-expressed DEGs after 3 h from the UV-B exposure.

Metabolic pathways associated with DEGs were retrieved using KEGG. In particular, KEGG analysis was performed on “phenylpropanoid biosynthesis”, “flavonoids biosynthesis”, and “circadian rhythm” maps (Figure 2a, Figure 2b, and Figure 2c, respectively), since the resulting metabolites from the aforementioned pathways were increased in UV-B-exposed peach fruits according to our previous metabolomic studies [12,29,30].

Concerning phenylpropanoids, we detected four and one over-regulated pathways after 1 and 3 h from the UV-B exposure, respectively. In particular, the gene encoding an O-hydroxycinnamoyltransferase (HCT; prupe.1g237100.1), with KO code K13065, resulted in being activated during both recovery time points (Figure 2a). The remaining four genes in the phenylpropanoid pathways, activated exclusively at the first time point (1 h), encoded the following enzymes: peroxidase (POD, prupe.7g016500.1), phenylalanine ammonia-lyase (PAL, prupe.6g235400.1), and caffeoylshikimate esterase (CSE; prupe.5g109300.1).

A flavonoid biosynthesis map showed three genes that were over-expressed after the UV-B treatment and shared after 1 and 3 h (Figure 2b). In particular, these genes encoded the CHS (prupe.i005700.1), the flavonol synthase (FLS, prupe.1g502800.1), and the HCT (prupe.1g237100.1; Figure 2b). Interestingly, the *CHI* (prupe.2g263900.1) gene was over-expressed in the flavonoid biosynthesis pathway exclusively after 3 h from the UV-B treatment (Figure 2b).

Finally, concerning the “circadian rhythm” map, we retrieved three genes activated by UV-B exposure at the two time points (Figure 2c). These genes encoded CHS (prupe.i005700.1), E3 ubiquitin-protein ligase RFWD2 (RFWD; prupe.2g293400.1), and transcription factor HY5 (prupe.1g208500.2).

For each analyzed map, no under-expressed genes were retrieved after the UV-B exposure.

### 2.3. Cellular Function Clustering of DEGs Exposed to UV-B

Possible cellular and molecular function of DEGs from the peel of peach fruits exposed to UV-B were clustered using MapMan. Especially, the map called “Overview” allowed the clustering of DEGs in cellular function such as transcription factor, cell wall, and external stimuli response (Figure 3).

Concerning transcription factors, we detected a major over-expression at the first time point with an overall 45 DEGs (39 OE, 6 UE), often belonging to the MYB, WRKY, and bZIP multigenic family (Appendix A). In addition, we also detected the *HY5* gene (Figure 3; prupe.1g208500.2). After 3 h from UV-B exposure, a total of six DEGs involved in gene regulation were retrieved; especially, two DEGs were shared with the first time point (MYB and DREB transcription factors) (Figure 3; Appendix A). Interestingly, we also retrieved two C2H2 zinc finger transcription factors, which correspond to genes *ZAT10* (prupe.1g424300.1) and *ZAT12* (prupe.2g230800.1).

Regarding cell wall, a total of six genes were regulated in peach fruit at the 1 h recovery time point. In particular, five genes resulted in overexpression, two *galacturonosyltransferases* (*GalAT*s; prupe.3g276500.1, prupe.1g384100.1), *xylan O-acetyltransferase* (*XOAT*; prupe.3g123000.1), *beta-D-xylosidase* (*BXL*; prupe.1g123100.1), and *CSE* (prupe.5g109300.1), whereas only one gene involved in cell wall formation, a *pectin methylesterase* (*PME*, prupe.7g190400.1), was under-expressed at the first time point of the experiment. No genes related to the cell wall were regulated at the second time point.

Finally, considering the map “external stimuli response”, upregulation of 10 genes occurred at the 1 h time point, three of which were activated also in the second time point (Figure 3). The genes that were overexpressed at both experimental times were *COP1* (prupe.4g013500.1), *RUP* (prupe.2g293400.1), and five genes belonging to multigenic families of *CBF/DREB1* (Figure 3; Appendix A). Furthermore, after 3 h from the treatment, 2 additional genes annotated as “UV-B signal transduction transcriptional regulator” were retrieved in the “external stimuli response map” (Figure 3). No under-regulated genes were observed within this map. A list of differentially expressed genes associated with the analyzed functional classes is available in Appendix A.

### 2.4. Co-Expression Network Analysis of DEGs

Possible gene network modules in the fruit peel of peach fruits exposed to UV-B radiation were identified by analyzing DEGs obtained at the first time point (after 1 h of recovery). These modules were also associated to biochemical values, such as flavonols, anthocyanins, lignans, and other phenolic classes (Appendix A).

Overall, 16 modules containing DEGs were retrieved and separated by color clusters (Appendix A). In particular, our analysis was focused on modules called “Red” and “Salmon”, since they include genes involved in the phenylpropanoid biosynthesis and UVR8-related pathways. “Red” and “Salmon” modules contained 35 and 19 genes, respectively.

These two modules were further analyzed to detect possible gene network associations (Figure 4). In the “Red” module, principal connections were shared by genes *CHS* (prupe.i005700.1), *MYB12* (prupe.8g270000.1), and *PAL* (prupe.6g235400.1; Figure 4A). Concerning the “Salmon” module, the main network was shared between the *endoplasmic reticulum*-localized *adenine nucleotide transporter 1* (*ER-ANT1*; prupe.5g208700.1), *HY5 homolog* (*HYH*; prupe.1g208500.1), and *DONGLE* (*DGL*; prupe.1g181800.1) transcripts (Figure 4B).

## 3. Discussion

Post-harvest UV-B radiation as a factor able to rearrange fruit transcriptome during storage has been scarcely investigated, since most current relevant literature focused on studying the effects of this radiation on fruit exposed in pre-harvest [31,32,33]. Considering the 847 DEGs found, almost all of them (88%) were detected after 1 h from the UV-B irradiation, with a predominance (78%) of the overexpressed ones compared to the under-expressed indicating that deep transcriptome modification can occur also during the first time of recovery. To date, no studies have investigated the timing of the UV-B-induced changes of gene expression in fruit during storage. However, in accordance with our findings, in *Arabidopsis*, a strong upregulation of some UVR8-related genes (*HY5*, *RUP1*, and *RUP2*) in plants irradiated with UV-B (3 μmol·m^−2^·s^−1^) was observed already after 1 h from the irradiation, suggesting a rapid activation of the UVR8 signaling pathway leading to an increase in the expression of downstream genes [34]. Additionally, the same study reported a decreasing trend in the transcript level of the same three genes after 3 h from the beginning of the irradiation. Another study on *Arabidopsis* investigating the photo-equilibrium between the dimeric (inactive) and monomeric (active) state of UVR8 exposed to 3·μmol m^−2^·s^−1^ UV-B found that the monomerization rate reached its maximum (25%) after 45–60 min from the beginning of the irradiation, further triggering the transcriptional changes in the downstream responsive genes [35].

UV-B radiation is known to activate the phenolic pathways through overexpression of specific biosynthetic and regulatory genes [7,36]. Most relevant studies in this field used *Arabidopsis* as the model plant [36,37,38], while the few pertinent manuscripts about UV-B-irradiated fruit [10,12,39,40,41] focused on specific putative UV-B-responsive phenylpropanoid genes rather than extensively investigating the changes in the transcriptome adopting an -omics approach. Overall, transcriptomic analyses showed an increased function of GOs such as metabolic and cellular processes, with a specific activation of genes related to “aromatic compound biosynthetic process” such as gene pathways involved in production of phenols. Interestingly, some metabolic pathways strongly influenced by UV-B treatment, according KEGG analysis, were “phenylpropanoid biosynthesis” and “flavonoids biosynthesis”. Particularly, a few of the upregulated genes belonged to the shikimate pathway, which represents the early stage of the phenolic biosynthesis. Most relevant literature focuses on the UV-B-mediated modulation of flavonoid-related genes, although some published studies show that also the early stages of the phenylpropanoid pathway are triggered by UV-B radiation [42,43]. More in detail, *PAL*, encoding a key enzyme in initiating the flavonoid biosynthesis, was found to be overexpressed after 1 h from the irradiation, indicating a UV-B-mediated stimulation of the phenolic pathway. *PAL* was indeed found to be a highly UV-B-regulated gene in some plant species [44,45,46], resulting in a higher enzymatic activity [45,46,47,48]. In addition to *PAL*, our results showed an overexpression of *HCT*, *CSE*, and *POD* in the 1 h recovery samples, which are crucial genes in the phenylpropanoid pathway. In detail, HCT catalyzes the conversion of p-coumaroyl CoA to *p*-coumaroyl shikimate but intervenes also in the reaction from caffeoyl shikimate to caffeoyl CoA [49]. Conversely, CSE esterifies the caffeoyl shikimate to produce caffeate [49], while POD plays a key role in catalyzing monolignol polymerization into lignin, besides acting as an antioxidant enzyme involved in ROS scavenging. These enzymes are crucial points in the biosynthesis of monolignols, which constitute the monomers of the lignin polymer [50]. It is well-known that, in higher plants, lignification represents a defensive response toward several biotic and abiotic stressors [49,50,51,52,53,54,55,56,57]. Since UV-B radiation constitutes a highly energetic portion of the solar spectrum, plants have evolved anatomical and morphological adaptations, in addition to biochemical and physiological modifications, to avoid damages to the photosynthetic apparatus and impairments in the biological processes [4,58]. Among them, a UV-B exposure has been found to increase not only the content of flavonoids but also the accumulation of lignin in the trichomes and cotyledons of different plant species [59,60,61,62]. The UV-B-triggered overexpression of genes involved in the lignin biosynthesis observed in our study might likely determine an increase in lignin production, attenuating the UV-B radiation as an acclimation response and inducing a higher resistance toward biotic stress [49,50,51,52,53,54,55,56,57,58,59,60,61,62,63]. In this regard, a more in-depth investigation of the UV-B-triggered rearrangement of the cell wall architecture, as well as the UV-B-induced modifications of lignin content and composition, will be the object of future work focused on these aspects.

According to the flavonoid biosynthetic pathway resulting from KEGG analysis, *CHS* and *FLS* were found to be upregulated after both 1 h and 3 h from the end of the UV-B exposure, while also *CHI* was overexpressed 3 h after the treatment. An increase in *CHS*, *CHI*, and *FLS* transcript levels has been observed in many UV-B-treated plants and fruit species, e.g., peaches [12,41,64], apple [10,65], tomato [39], and table grape [66]. UV-B-triggered promotion of flavonoid-related genes has been described as an acclimation response toward UV-B conditions [5,36,67], since the antioxidant capacity of flavonoid molecules can effectively neutralize the UV-B-induced ROS [68,69], preventing damages to cellular components, such as nucleic acids and proteins. Our study also showed no under-expressed genes belonging to either phenylpropanoid or flavonoid biosynthesis in any of the recovery time points (1 h and 3 h), indicating that only an upregulation in the phenolic pathway occurred following the UV-B exposure.

Although UV-B perception and signaling via UVR8 have been deeply studied in *Arabidopsis*, only a few studies investigated the activation of UVR8-pathway-related genes in fruits [12,41,70,71,72]. Interestingly, considering the “circadian rhythm” map resulting from the KEGG analysis, and the DEGs classification through MapMan, an upregulation of *HY5*, *COP1*, and *RUP* genes was found in both the recovery time points, 1 h and 3 h. Transcription of *HY5* and *RUP* genes is UV-B-regulated, since they play a key role in activating the expression of specific UVR8-responsive genes and promoting the UVR8 re-dimerization, respectively [73,74,75,76]. Upregulation of *HY5* and *COP1* was also observed in UV-B-treated peach peel and pulp [12,41], although the authors investigated the transcript levels not earlier than 6 h after the end of the UV-B irradiation. The present study provides evidence that upregulation of *HY5* and *COP1* in the peel of UV-B-exposed peach fruit, likely because of the activation of the UVR8 receptor, occurred already after 1 h from the end of exposure. Indeed, as aforementioned, a time-course experiment on *Arabidopsis* shows that, after exposing the plants to broadband UV-B radiation (3 μmol·m^−2^·s^−1^), 1 h of recovery is sufficient to get the highest (~25%) UVR8 monomerization rate [35]. The much greater activation of genes encoding for UVR8- and phenylpropanoid-related pathways detected at 1 h recovery, compared to the 3 h recovery, suggests that the UV-B signal is weakening at the second time point considered. However, combining this observation with the one by Santin et al. [12], who found a biphasic activation of *HY5* and *COP1* after 6 and 24 h from the irradiation, it can be hypothesized that the transcript levels of such genes might undergo fluctuations over time after their activation.

A MapMan-based cluster of DEGs revealed also transcriptional modifications of enzymes involved in cell wall plasticity. In particular, after 1 h from the UV-B irradiation, a down-regulation of a PME isoform was found. PMEs constitute a multigene family, accounting for more than 66 members in *Arabidopsis* [47], 47 in *Vitis vinifera* [77], 60–70 in *Musa acuminata* [78], 78 in *Gossypium raimondii* [79], 54 in *Fragraria vesca* [80], 79 in *Solanum lycopersicum* [81], and 71 in *Prunus persica* genome [82]. PMEs act by promoting the demethylesterification of the homogalacturonan polymer, contributing to the cell wall disassembly [83,84]. Therefore, PMEs, together with other cell-wall-dismantling enzymes such as polygalacturonases and β-galactosidases, play a crucial role in the softening process during fruit ripening and senescence, contributing to the overall commercial and organoleptic quality of the products. A UV-B-induced decrease in PME activity in the peel of peach fruit was previously observed [85,86]. Indeed, the authors found a reduction in PME activity 12 and 36 h after a 60 min UV-B exposure, while a decreased *PME1* expression was detected 12 and 24 h after the UV-B treatment. In the present study, a downregulation of the *PME* gene has been found already after 1 h from the end of the UV-B exposure, confirming the influence of UV-B wavelengths in affecting *PME* expression. In any case, a decrease in *PME* expression, accompanied with reduced PME activity, might pave the way for the use of UV-B irradiation to extend the shelf life of fruit and vegetable products. Additionally, the current study revealed an increase in other genes encoding for enzymes involved in cell wall organization, such as two GalATs (fold changes 1.36 and 1.76) and a XOAT (fold change 1.21), after 1 h from the end of the UV-B irradiation. GalATs and XOATs are two key enzymes in contributing to the functional architecture and the mechanical properties of the cell wall, by catalyzing the addition of *O*-acetyl residues to the xylan and by adding galacturonic acid moieties to form pectin polymer, respectively [87,88].

To detect a correlation between UV-B-triggered transcriptomic and metabolomic changes, a co-expression network analysis of DEGs has been conducted, and the resulting networks were associated with the metabolomic dataset of a previous manuscript from the same authors [12]. This analysis was performed using the transcriptomic dataset from the 1 h recovery time point, since most of the UV-B-induced changes were observed after 1 h from the irradiation (88% of DEGs), and the number of upregulated phenylpropanoid-related genes was higher compared to the 3 h recovery. One of the most interesting DEGs-containing clusters, the “Red” one, included some upregulated genes involved in the phenylpropanoid pathway, such as *CHS* and *PAL*, and the gene encoding for the transcription factor MYB12, which was found to promote the transcription of some flavonoid-related genes, such as *CHS* and *FLS* [89,90,91]. The “Salmon” cluster contained the *HYH* gene, whose transcription is activated by UV-B radiation via UVR8 and, partially overlapping with HY5 transcription factor, acts by promoting the expression of downstream genes involved in UV-B acclimation and photoprotection [36,92,93,94]. A strong correlation was found between genes belonging to the “Red” and “Salmon” clusters and the accumulation of specific phenolic subclasses, e.g., anthocyanins, flavones, dihydroflavonols, and flavanones, which are among the phenolic groups exhibiting the greatest antioxidant capacity. Interestingly, the gene network associations analysis (Figure 4) revealed that the principal connections resulting from the “Red” module were shared by two biosynthetic (*CHS* and *PAL*) and a regulatory (*MYB12*) gene belonging to the phenylpropanoid pathway. According to the model proposed by Santin et al. [12], the overall accumulation of phenolic compounds, especially flavonoids, detected after 36 h from the UV-B treatment could be the result of the UVR8-triggered activation of genes involved in the phenylpropanoid pathway. Through qRT-PCR, the authors observed an upregulation of specific biosynthetic (*CHS*, *F3H*, *F3′H*, and *DFR*) and regulatory (*MYB111* and *MYB-like*) genes, as well as genes involved in the UVR8 pathway (*COP1* and *HY5*) after 6 h from the exposure, but earlier recovery time points were not investigated in that study. With this work, through an -omics approach, we found that the activation of the UVR8 pathway occurred already after 1 h from the end of the UV-B treatment and continues also 3 h after, since *HY5* and *COP1* were upregulated in both the time points considered. Interestingly, also some downstream flavonoid-related genes were activated both after 1 (*PAL*, *CHS*, *FLS*) and 3 h (*CHS*, *CHI*, *FLS*). As a result, the accumulation of specific phenolic subclasses was visible 36 h after the exposure [12], indicating a UV-B-triggered accumulation of such health-promoting compounds through transcriptional activation. However, it is important to state that the UV-B triggered accumulation of phenolic compounds is strictly dependent on both genotypes, influencing the biochemical changes to external factors such as a UV-B irradiation, and the UV-B dose to which the plant material is exposed [21]. Indeed, considering the importance of the cultivars, different responses to the same UV-B treatment were observed in post-harvest peaches [28] but also in other plant species, such as lettuce [95], blueberry leaves [96], and olives [97]. Besides, the UV-B treatment conditions, in terms of duration and irradiance, are also crucial to determine specific biochemical modifications in the UV-B-exposed plants of fruits. Indeed, experiments conducted on the same plant species and cultivar but varying the UV-B dose gave significantly different results, such as in peach fruit [29] but also in other plant species, such as mung bean sprouts [47] and basil [98]. In light of the aforementioned considerations, it is important to highlight that the observed responses in terms of both gene expression and correlation with metabolite accumulation are strictly cultivar- and UV-B-dose dependent; therefore, they might significantly differ if a different peach cultivar and/or different UV-B treatment conditions are considered.

A suggested working model of the transcriptional modifications and the resulting accumulation of specific phenolic classes in the peel of UV-B-treated peach fruit is presented in Figure 5.

Finally, transcription factors differentially regulated in the peel of peach fruit exposed to UV-B were collected in our analysis. In addition to the HY5 and MYB factors previously described, we retrieved several genes encoding for multigenic families involved in transcriptional regulation such as *bHLH*, *DREB*, *C2H2 zinc finger*, *ERF*, *GATA*, *GRAS*, and *WRKY* (Figure 3; Appendix A). The bHLH transcription factors family has been observed to respond to environmental UV-B signals, cooperating with other TFs, for example HFR1, stabilizing the cell light response via the UVR8 pathway [2,99]. DREB and C2H2 zinc finger multigenic TFs are positively regulated in response to abiotic stress and also during UV-B irradiation [92]; interestingly, in our analysis, we detected the activation of two genes belonging to C2H2 transcription factors, corresponding to *ZAT10* and *ZAT12*, the expressions of which have been observed in *A. thaliana* to rapidly increase after 1 h from UV-B exposure [100]. Irradiation by UV-B in plant leaves is also known to stimulate ethylene production, as in the case of *A. thaliana* and grapes, by inducing the activation of the ERF transcription factor [101,102]. Some members of this multigenic family were retrieved as overexpressed during peach peel exposure to UV (Appendix A). Concerning GATA and GRAS transcription factors, recent studies pointed out the activation of these genic families during the circadian rhythm; especially the GATA box can physically interact with HY5 to regulate response to UV-B radiation [103,104]. Regarding WRKY, these transcription factors have been described as activated concomitantly to UV-B stimulation in rice, enhancing tolerance of leaf surface to irradiation by increasing lignin [105].

## 4. Materials and Methods

### 4.1. Plant Material and UV-B Treatment

Organic peach fruits (*Prunus persica* L., cv Fairtime), uniform in size (8.1 cm average diameter) and color (mostly yellow with few orange spots) and without any wound or damage, came from the farm Azienda Agricola Conforti Cristiano (43°53′42.867″ N, 10°30′57.178″ E , 48 m above sea level) and were treated with UV-B radiation within 24 h after harvesting. Once at the laboratory, peaches showed a firmness value of 25.60 ± 0.18 N, defining the stage of the fruit as “ready to buy” [106]. The fruits were randomly assigned to either control (n = 10) or UV-B-treated (n = 10) group. The UV-B treatment was conducted at room temperature (24 °C) using UV-B tubes (Philips Ultraviolet-B Narrowband, TL 20 W/01—RS, Koninklijke Philips Electronics, Eindhoven, the Netherlands) together with white light tubes (Philips F17T8/TL741). Control fruits were placed in a separated climate chamber at the same temperature conditions but exposed to just white light. The UV-B exposure lasted 60 min, corresponding to a total irradiance of 38.53·kJ·m^−2^ (8.33 kJ·m^−2^ UV-B + 30.20 kJ·m^−2^ white light), while control fruits were given a total irradiance of 30.20 kJ·m^−2^ white light. The UV-B dose and experimental conditions were chosen according to previous studies by the same research group [12,29]. Randomized groups of five peaches from both the control and the UV-B-treated groups were sampled after 1 and 3 h from the end of the UV-B treatment. During the recovery period, the fruits were kept at the same climate chambers at room temperature (24 °C) with just the white light on. Then, the peach peel from the UV-B-exposed side of the fruit was accurately removed with scalpels and tweezers, immediately dipped into liquid nitrogen, and kept at −80 °C until analysis.

### 4.2. RNA Isolation

Total RNA was extracted with the LiCl/CTAB method with few changes, as reported in detail in a previous manuscript by the same authors [85]. Briefly, samples were ground in liquid nitrogen and extraction buffer (2% [*w*/*v*] hexadecyltrimethylammonium bromide, CTAB; 2% [*w*/*v*] polyvinylpyrrolidone (average molecular weight 40,000), PVP; 100 mM Tris/HCl pH 8.0; 25 mM EDTA; 2 M NaCl; 0.5 g·L^−1^ spermidine and 2.7% [*v*/*v*] 2-mercaptoethanol) was added to the freeze material. Ice-cold chloroform:isoamyl alcohol (24:1) was used for phase separation. Addition of 10 M LiCl at 4 °C with following overnight incubation allowed RNA-selective precipitation. After rinsing with cold 70% EtOH and re-hydration in 100 µL RNAse-free water, the RNA samples were stored at −80 °C. Purification from genomic DNA was performed by digestion with DNaseI (Roche, Basel, Switzerland). Finally, RNA was purified with phenol/chloroform and precipitated with standard procedures. RNA concentration and quality were checked through Qubit RNA BR Assay Kit (Invitrogen, Waltham, MA, USA), WPA Biowave spectrophotometer (Biochrom Ltd., Cambridge, UK), agarose gel electrophoresis, and bioanalyzer analysis using a Bioanalyzer 2100 (Agilent Technologies, Santa Clara, CA, USA).

### 4.3. cDNA Libraries Preparation

Libraries of cDNA deriving from the peels of control and UV-B-exposed fruits of *P. persica* considering two recovery time points (1 and 3 h after the irradiation) were collected and sequenced with an Illumina HiSeq 2500. RNA-seq data were deposited in the NCBI Sequence Read Archive (SRA) under accession number PRJNA855351. Three biological replicates randomly selected out of five fruits were collected for each treatment and time point. Overall quality of reads was checked by using FastQC (v.0.11.9) and improved using Trimmomatic (v. 0.3.9) [107] with the following parameters: SLIDINGWINDOW: 4:20; CROP: 80; HEADCROP: 10; MINLEN: 70. Possible ribosomal contamination was removed from high-quality cDNA libraries using CLC Genomic Workbench v. 9.5.3 (CLC-BIO, Aarhus, Denmark) and mapping reads on *P. persica* rRNA, downloaded from the SILVA [108] repository, with the following parameters: mismatch cost = 2, insertion/deletion cost = 3, length fraction = 0.5, and similarity fraction = 0.8. Not-mapped reads were retained for subsequent analysis.

### 4.4. Differential Expression Analysis

High-quality reads were aligned on the *P. persica* transcriptome [23] “https://phytozome-next.jgi.doe.gov/info/Ppersica_v2_1” (accessed on 15 October 2021) by using the CLC Genomic Workbench as follows: mismatch cost = 2; insertion/deletion cost = 3, length fraction = 0.8, and similarity fraction = 0.8. Raw counts per transcript were analyzed using the R package edgeR [109] and gene expression calculated as reads per kilobase per million reads mapped (RPKM) [110]. Genes with RPKM > 1 in at least one library were considered as expressed and used for further analysis. Differentially expressed genes (DEGs) were obtained using the likelihood test on edgeR performing pairwise comparison between control and UV-B-treated fruits during two time points (1 and 3 h, respectively). Genes were selected as differentially expressed for absolute fold change >2 and false discovery rate (FDR) [111] corrected *p*-value < 0.05.

### 4.5. Gene Ontology (GO) and DEGs Functional Classification

Description of genes and associated GO terms were downloaded from the phytozome database of *P. persica*. GO distribution for differentially expressed genes was visualized using WEGO with default parameters [112].

GO enrichment analysis was performed between GO terms of differentially expressed genes and GO terms of the whole transcriptome of *P. persica*. Statistical analyses were carried out with Fisher exact test using Blast2GO [113] and GO terms showing FDR-corrected *p*-value < 0.05 were considered enriched. Subsequently, REVIGO was used to remove redundant GO terms with the parameter “tiny similarity” [114].

MapMan was used for functional clustering and visualization of DEGs [115], correspondent MapMan bins of *P. persica* were retrieved using Mercator4 v. 2.0 [116] on protein sequences.

Finally, the KEGG database (Kyoto Encyclopaedia of Genes and Genomes) was used to retrieve metabolic pathways associated with DEGs. In particular, KEGG Orthology (KO) id codes associated to DEGs were obtained using protein sequences on KAAS [117] with the bi-directional blast hit (BBH) option. Subsequently, KO codes belonging to DEGs were submitted to the KEGG [111] mapper for metabolic pathway retrieval.

### 4.6. Gene Co-Expression Network Analysis and Metabolic Data Correlation

Expression values (RPKM) of DEGs retrieved by RNA-seq analysis during the first time point (1 h of recovery) were further investigated by the WGCNA algorithm [118] to construct co-expression network modules related to UV-B exposure. The soft threshold values were implemented and detected according to WGCNA instruction. In particular, the gradient method was used to test different power values, ranging from 12 to 40, allowing us to establish the most suitable soft power value of 30 corresponding to an R2 threshold of 0.85. The minimum number of genes per module was set to 15.

Correlation analysis between eigengenes from each module and the biochemical values of phenolic classes (Appendix A) was performed by using WGCNA to assess statistical linear regression with corresponding *p*-value for each module. The phenolic data used for correlation analysis was derived from an extensive re-processing of metabolomic data derived from a previous experiment [29], conducted on the same peach cultivar adopting the same experimental conditions and UV-B treatment. Correlation with the quantification of several individual phenolic subclasses (expressed as the sum of integrated peak areas of each phenolic subclass components; Appendix A) was performed considering the data from the 36 h recovery time point, since the accumulation of specific phenolic subclasses was observed only 36 h after the end of the UV-B exposure, likely because of the antecedent UVR8-mediated activation of flavonoid biosynthetic genes [12].

Network visualization for key genes corresponding to co-expression network modules was performed using VisAnt [119] with the cut-off weight parameter set to 0.3. In the representation, each node corresponds to a gene that is connected to a different number of genes by lines.

## 5. Conclusions

To conclude, although some studies have investigated the UV-B-induced upregulation of phenylpropanoid genes in different fruit species, no previous works focused on the transcriptomic modifications in the peel of UV-B-exposed peach fruit and their correlation with the content of phenolic compounds. This study provides evidence that the UV-B exposure triggers the UVR8 pathway also in peach fruit by upregulating some well-known UVR8-related genes, such as *HY5* and *COP1*, whose increase in transcript levels was visible already after 1 h from the irradiation and continues after 3 h. As a result, a downstream activation of some phenylpropanoid biosynthetic and regulatory genes occurred, determining a consequently strong increase in specific phenolic subclasses. In conclusion, this manuscript deepens the knowledge of the UV-B-related transcriptomics changes in fruit, so far widely explored only in *Arabidopsis*.

## Figures and Tables

**Figure 1 plants-12-01818-f001:**
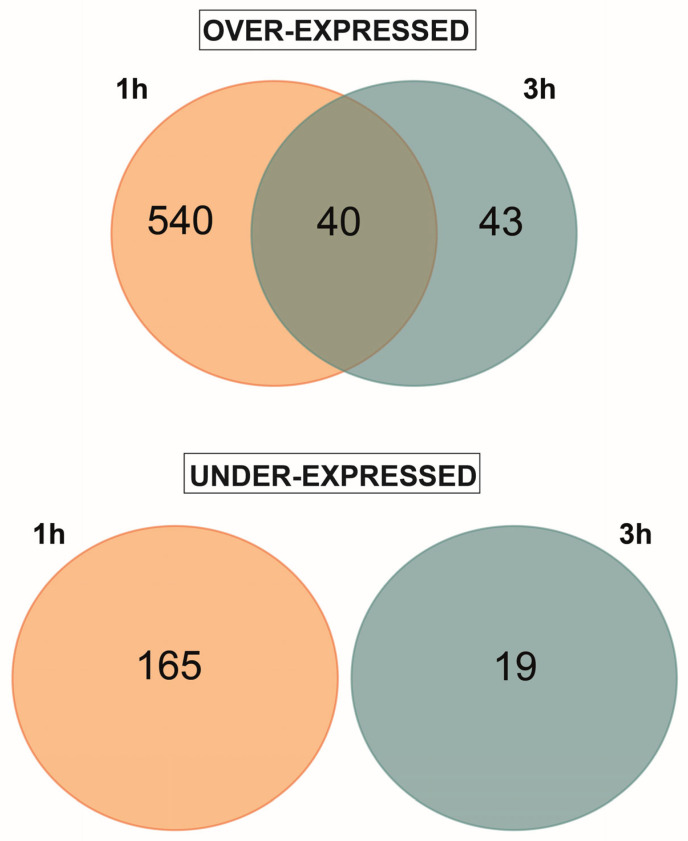
Venn diagram for over- and under-expressed genes detected in the peel of UV-B-treated peach fruit after 1 h and 3 h from the end of UVB exposure.

**Figure 2 plants-12-01818-f002:**
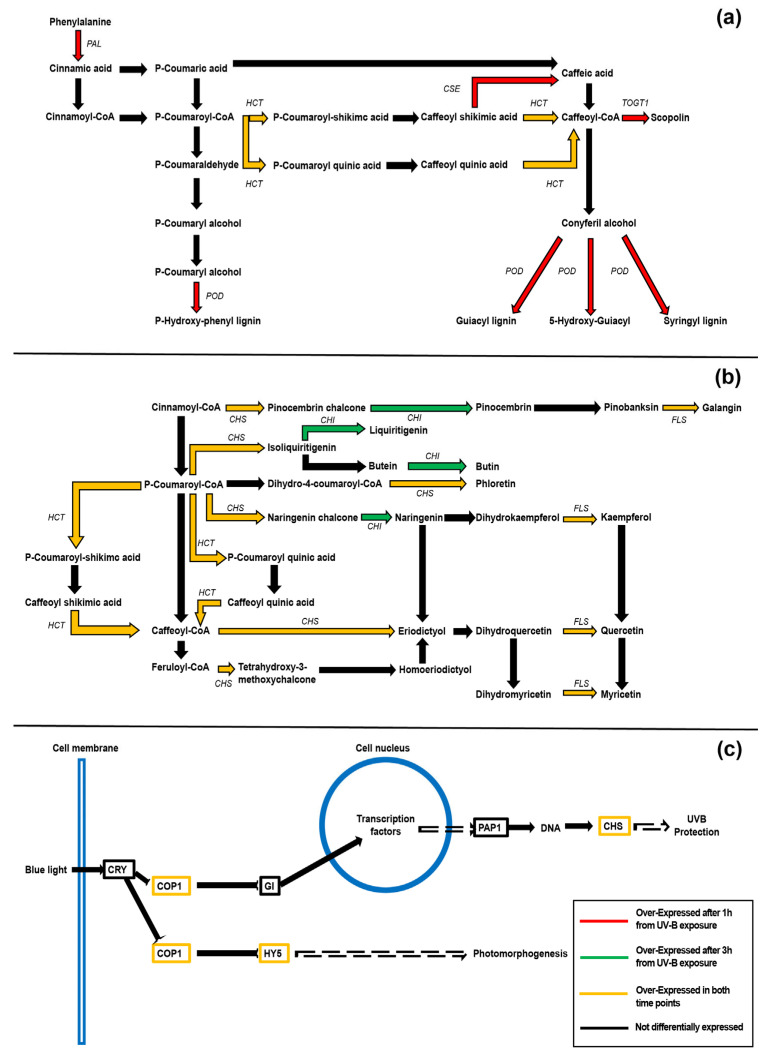
Schematization of metabolic pathway for phenylpropanoid (**a**), flavonoid (**b**), and circadian rhythm (**c**), as described by KEGG 111 after 1 h and 3 h from the end of UVB exposure in the peel of UV-B-treated peach fruit. Arrows indicate the path from reagent to product. Red arrows underline genes encoding for enzyme that resulted in being over-expressed after 1 h of recovery; similarly, green arrows underline over-expressed genes after 3 h of recovery; yellow arrows are over-expressed DEGs active during both recovery times. Black arrows are for not-differentially expressed genes in the related pathways. No under-expressed genes were retrieved for the investigated pathways. Genes involved in the metabolic pathway are written in italics. Concerning the circadian rhythm map (**c**), genes involved in cellular regulation are within the square.

**Figure 3 plants-12-01818-f003:**
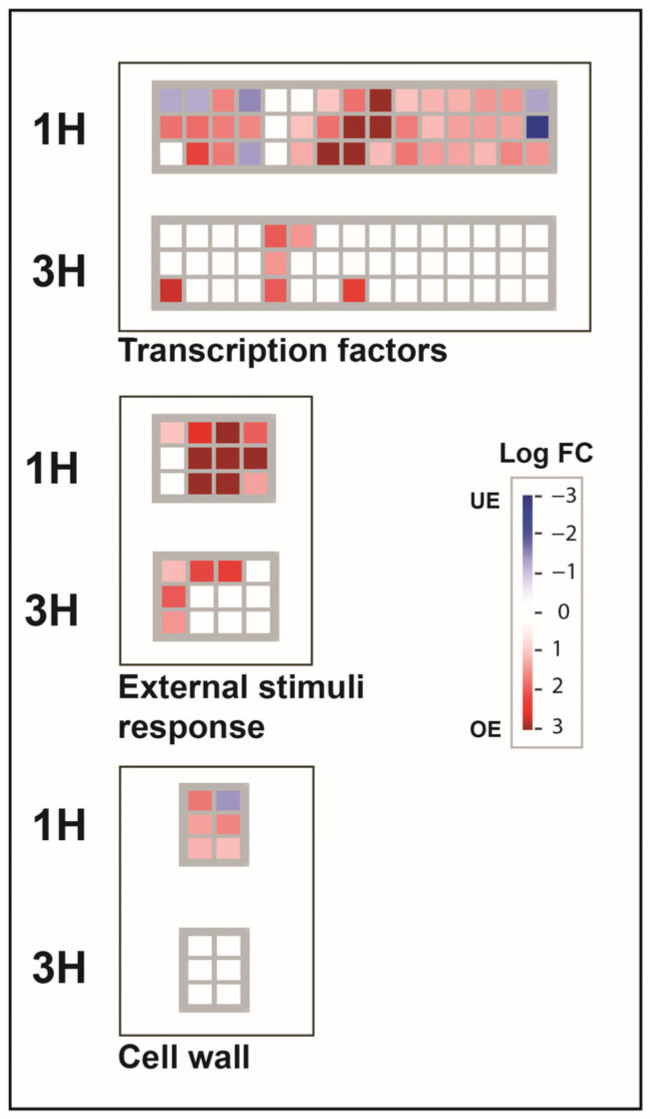
MapMan schematization for DEGs found in the peel of UV-B-treated peach fruit after 1 and 3 h of recovery. Red dots represent the over-expressed genes (OE), whereas blue dots are under-expressed ones (UE). White dots indicate genes that were not differentially expressed in one time point but in the other. The scale, based on gene fold change, spans from dark blue (Log FC = −3) to dark red (Log FC = 3).

**Figure 4 plants-12-01818-f004:**
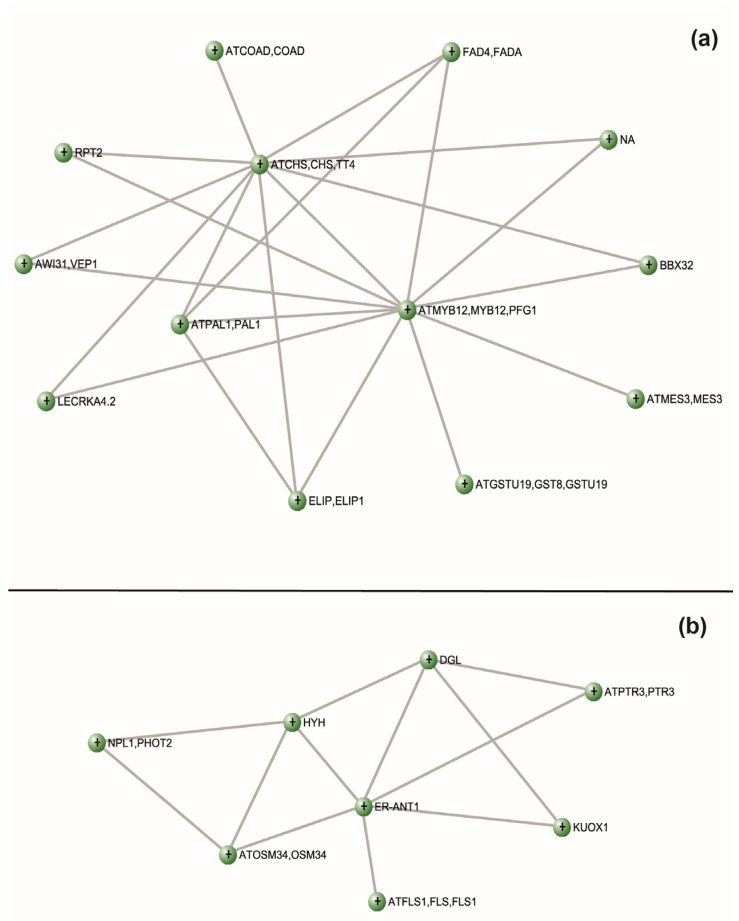
VisAnt gene network of (**a**) “Red” and (**b**) “Salmon” co-expression modules detected with WGNA. Each node represents a gene, interactions between nodes are shown as lines.

**Figure 5 plants-12-01818-f005:**
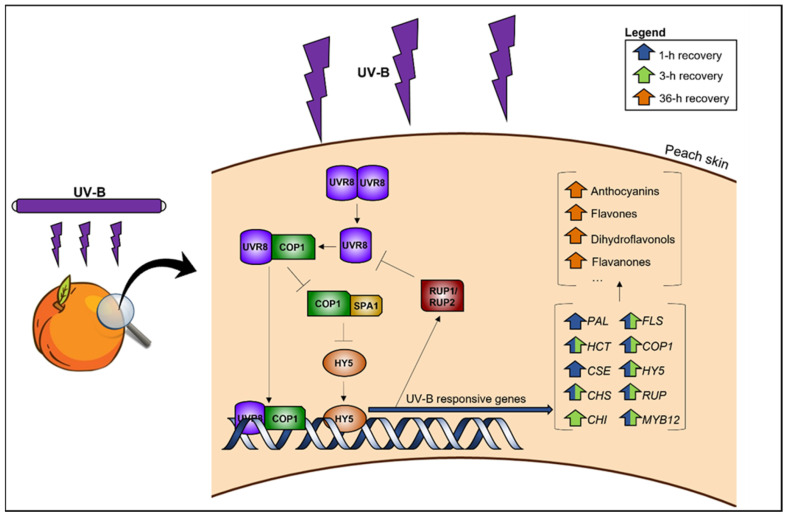
Summary of the molecular and biochemical responses of post-harvest peach fruit irradiated with UV-B and the UVR8 pathway model [3].

**Table 1 plants-12-01818-t001:** Summary statistics for the Illumina sequencing and mapping against *Prunus persica* reference transcriptome. CTR = control; UV-B = UV-B treated; 1 h = 1 h of recovery after the UV-B treatment; 3 h = 3 h of recovery after the UV-B treatment.

Library	Replicate	Raw Reads Per Library (N)	Trimmed Reads Per Library (N)	Aligned Reads on *P. persica* Reference Transcriptome (N)	Aligned Reads on *P. persica* Reference Transcriptome (%)
CTR_1 h	1	33,257,571	32,949,871	31,328,689	95.08
	2	28,285,172	28,032,925	26,375,914	94.09
	3	27,168,547	26,980,657	25,341,162	93.92
UV-B_1 h	1	27,439,039	27,181,626	25,618,232	94.25
	2	27,978,969	27,717,323	26,033,386	93.92
	3	30,422,616	30,161,462	28,388,389	94.12
CTR_3 h	1	26,980,674	26,692,135	25,040,323	93.81
	2	28,983,969	28,740,615	26,775,717	93.16
	3	24,796,737	24,601,286	22,984,308	93.43
UV-B_3 h	1	28,917,100	28,587,951	27,065,772	94.68
	2	31,460,791	31,111,037	29,191,493	93.83
	3	23,325,335	23,126,890	21,689,653	93.79

## Data Availability

Libraries of cDNA were deposited at the NCBI sequence reads archive (SRA; https://www.ncbi.nlm.nih.gov/sra (accessed on 30 July 2022) under bioproject accession number PRJNA855351.

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
