# Peer review of "Transcriptomic Analysis on the Peel of UV-B-Exposed Peach Fruit Reveals an Upregulation of Phenolic- and UVR8-Related Pathways"

_plants, 2023, doi:10.3390/plants12091818_

Round 1

Reviewer 1 Report

This manuscript has serious flaws. More experiments need to be added. The data are not enough to support a research article. The author is advised to replan the experimental design and add more significant results to this research article in order for it to be suitable for publication-

Author Response

Reviewer 1

This manuscript has serious flaws. More experiments need to be added. The data are not enough to support a research article. The author is advised to replan the experimental design and add more significant results to this research article in order for it to be suitable for publication-

Response from the authors: We are sorry that the Reviewer finds our manuscript not suitable for publication, since we do believe that the experimental design adopted, the materials and methods used, and the results obtained meet the rigorous scientific quality standards needed for a scientific article. In fact, three out of four Reviewers have judged our manuscript positively.

Reviewer 2 Report

I have made a fair number of comments/raised questions within the attached copy of your manuscript. Perhaps the general themes of my comments are:  i) your figures do, in my opinion, an exceptionally good job of bringing out the main points of your research--especially the closing model, and ii) I am left wanting some clarification on the practical significance of this work (that is, does your work support the use of UV-B as a mechanism to nutritionally enhance peach fruit after harvest?

Author Response

Reviewer 2

I have made a fair number of comments/raised questions within the attached copy of your manuscript. Perhaps the general themes of my comments are:  i) your figures do, in my opinion, an exceptionally good job of bringing out the main points of your research--especially the closing model, and ii) I am left wanting some clarification on the practical significance of this work (that is, does your work support the use of UV-B as a mechanism to nutritionally enhance peach fruit after harvest?

Response from the authors: We thank the Reviewer for his/her positive evaluation of our manuscript.

Regarding the comment on the practical significance of this study, the present work is part of a research regarding the effect of post-harvest UV-B exposure on peach fruit (both peel and pulp) at biochemical and molecular level. As the Reviewer pointed out, the final and practical aim is undoubtedly to enhance the nutraceutical quality of UV-B-irradiated peach fruit by enhancing the content of health-promoting secondary metabolites. The present paper aims to investigate deeper and comprehensively (through an -omics approach) the molecular mechanisms behind the biochemical modifications observed previously, since the practical aspects of the whole research are better defined in the more proper manuscripts already published (see the reference list below). However, as indicated by the Reviewer’s comment in the attached pdf file, we have added the following sentence to the abstract to better frame the practical aim of the study: “Thus, our results might pave the way to a possible use of post-harvest UV-B treatments to enhance to content of health-promoting compounds in peach fruits and extending the knowledge of UVR8 gene network.”

Santin, M., Ranieri, A., Hauser, M. T., Miras-Moreno, B., Rocchetti, G., Lucini, L., ... & Castagna, A. (2021). The outer influences the inner: Postharvest UV-B irradiation modulates peach flesh metabolome although shielded by the skin. Food chemistry, 338, 127782.

Santin, M., Lucini, L., Castagna, A., Chiodelli, G., Hauser, M. T., & Ranieri, A. (2018). Post-harvest UV-B radiation modulates metabolite profile in peach fruit. Postharvest Biology and Technology, 139, 127-134.

Santin, M., Castagna, A., Miras-Moreno, B., Rocchetti, G., Lucini, L., Hauser, M. T., & Ranieri, A. (2020). Beyond the visible and below the peel: How UV-B radiation influences the phenolic profile in the pulp of peach fruit. A biochemical and molecular study. Frontiers in Plant Science, 11, 579063.

Santin, M., Lucini, L., Castagna, A., Rocchetti, G., Hauser, M. T., & Ranieri, A. (2019). Comparative “phenol-omics” and gene expression analyses in peach (Prunus persica) skin in response to different postharvest UV-B treatments. Plant physiology and biochemistry, 135, 511-519.

We have revised the text according to all his/her observations in the attached pdf file. Particularly, we answer the Reviewer’s comments that need some clarifications below.

For my understanding of English, this phrase should be "fruits and vegetables". Also later in this paragraph.

Response from the authors:  Reviewer is right for what concerns “vegetable” noun. However, "fruit" is considered as an uncountable noun when is referred to fruit as a category of foods (as in the Introduction section). Indeed, the instructions for authors of some scientific journals clearly indicate to write “fruit” always without the final “s”. However, we have carefully checked this throughout the text.

now, I think that we are back to my original question in the abstract--sounds like people are using UVB to obtain desireably higher levels of flavonoids in fruits and vegetables. Seems that I cannot get this straight.

Response from the authors: The Reviewer is right. Since few years, the attention towards the environment has led researchers to find new, green, and sustainable solutions to improve the quality of fruit and vegetables. However, we do mean only in the research field, since more studies are needed to transfer this eco-friendly technology to a larger scale (growers, producers, or industries).

Will you tell your reader if 94% aligned reads is a favorable outcome, vis-a-vis other similar projects?

Response from the authors: Thanks to the reviewer for the comment. Yes, 94% of aligned reads in an RNA-seq experiment is a favourable outcome, as Conesa et al. (2016) pointed out, outlining a suitable range or 70-90% of mapped reads for a reliable gene expression analysis. Furthermore, many other articles regarding RNA-seq analysis using the genome of the species as a reference for mapping reads showed similar results, such as Sun et al. (2021). The text has been changed accordingly.

Conesa, A., Madrigal, P., Tarazona, S., Gomez-Cabrero, D., Cervera, A., McPherson, A., ... & Mortazavi, A. (2016). A survey of best practices for RNA-seq data analysis. Genome biology, 17(1), 1-19. 

Sun, Y., Zhang, T., Xu, X., Yang, Y., Tong, H., Mur, L. A. J., & Yuan, H. (2021). Transcriptomic characterization of nitrate-enhanced stevioside glycoside synthesis in stevia (Stevia rebaudiana) bertoni. International Journal of Molecular Sciences, 22(16), 8549.

Are you able/inclined to draw any inferences here about the use of UVB as a mechanism for enhancing flavonoid content for nutritional purposes?

Response from the authors: as suggested previously by the Reviewer, we have added in the Abstract section a short sentence regarding the possible use of UV-B as a sustainable tool to enhance the flavonoid content in post-harvest fruits. However, since this study is mainly focused on the transcriptomic responses to UV-B, and not on the metabolomic changes (apart from the correlation analysis between molecular and biochemical responses), we think that it might not be fully appropriate to insert a comment like this also in the Conclusion section, since it was not the scope of this specific work. As stated above, the inferences about the application of UV-B exposure to increase the content of health-promoting compounds has been widely described in other papers of the same research group.

Helpful figures; I do wonder if the under-expressed figure could/should have two separated circles, as there were zero genes in common across the two sampling times.

Response from the authors: We agree with the Reviewer, figure 1 has been changed according to suggestion

Reviewer 3 Report

Manuscript review "Transcriptomic analysis on the peel of UV-B-exposed peach fruit reveals an upregulation of phenolic- and UVR8-related pathways" by M. Santin et al.

Judging by the list of references, this work is a continuation of the authors' research on the effect of ultraviolet radiation on plants. The methods are adequate, the work is logically built correctly, a large array of data has been obtained, the correct conclusions and conclusions have been drawn. There are several wishes and a question.

L. 20. It is necessary to give the Latin name of the peach species (Prunus persica L., cv Fairtime).

L. 22. It is necessary to decipher the abbreviations in the annotation - PAL, CHS, CHI and FLS.

L.426. The authors write “…radiation within 24 hours”, and on L.433 “The UV-B exposure lasted 60 min”. This is a contradiction, where is the truth? Why treated 24 hours with UV-B after harvest?

What is the scientific novelty and practical significance of the work if the authors write that “In the light of the aforementioned considerations, it is important to highlight that the observed responses in terms of both gene expression and correlation with metabolites accumulations are strictly cultivar- and UV-B -dose dependent, therefore might significantly differ if a different peach cultivar and/or different UV-B treatment conditions are considered." This should be emphasized in the work.

Author Response

Reviewer3

Manuscript review "Transcriptomic analysis on the peel of UV-B-exposed peach fruit reveals an upregulation of phenolic- and UVR8-related pathways" by M. Santin et al.

Judging by the list of references, this work is a continuation of the authors' research on the effect of ultraviolet radiation on plants. The methods are adequate, the work is logically built correctly, a large array of data has been obtained, the correct conclusions and conclusions have been drawn. There are several wishes and a question.

Response from the authors: we thank the Reviewer for his/her positive evaluation of our manuscript.

L.20. It is necessary to give the Latin name of the peach species (Prunus persica L., cv Fairtime).

Response from the authors: We agree with the Reviewer, and we have added this information.

L.22. It is necessary to decipher the abbreviations in the annotation - PAL, CHS, CHI and FLS.

Response from the authors: Full names of the genes have been added accordingly.

L.426. The authors write “…radiation within 24 hours”, and on L.433 “The UV-B exposure lasted 60 min”. This is a contradiction, where is the truth? Why treated 24 hours with UV-B after harvest?

Response from the authors: We thank the Reviewer to giving the possibility to clarify. There is no contradiction, since we meant that the peaches were treated with UV-B (60 min irradiation time) within the 24 hours after harvesting. We wanted to conduct the irradiation as soon as possible after harvest, since peaches are highly perishable fruits, and a longer storage period (> 24 h) before the UV-B exposure would have needed proper storage parameters (e.g., low temperatures, RH conditions) that might have masked the effects due to the UV-B irradiation.

What is the scientific novelty and practical significance of the work if the authors write that “In the light of the aforementioned considerations, it is important to highlight that the observed responses in terms of both gene expression and correlation with metabolites accumulations are strictly cultivar- and UV-B -dose dependent, therefore might significantly differ if a different peach cultivar and/or different UV-B treatment conditions are considered." This should be emphasized in the work.

Response from the authors: The current relevant literature on UV-B-effects both at molecular and biochemical levels on post-harvest fruits have underlined that the responses (in terms of metabolites accumulation) can vary, since are dependent on the plant genotype and therefore its predisposition to accumulate such health-promoting compounds, which is connected to the plants’ environmental conditions. It has also been proved that the UV-B radiation has a great potential to enhance the content of such phytochemicals, and therefore it attracts the interest of a large portion of the scientific community. By far, what we know is that, if the positive effects of UV-B will be further confirmed, a protocol for the UV-B irradiation should be drawn up for each plant species and cultivar, similarly to what is currently done for the PAR regiment during indoor plant cultivation. Finally, in the light of this, the practical significance of the UV-B irradiance in terms of enhancing the nutraceutical quality of peach fruit has been widely described in more proper manuscripts of the same research group, that investigated specifically the changes in flavonoid content (see the reference below).

Santin, M., Ranieri, A., Hauser, M. T., Miras-Moreno, B., Rocchetti, G., Lucini, L., ... & Castagna, A. (2021). The outer influences the inner: Postharvest UV-B irradiation modulates peach flesh metabolome although shielded by the skin. Food chemistry, 338, 127782.

Santin, M., Lucini, L., Castagna, A., Chiodelli, G., Hauser, M. T., & Ranieri, A. (2018). Post-harvest UV-B radiation modulates metabolite profile in peach fruit. Postharvest Biology and Technology, 139, 127-134.

Santin, M., Castagna, A., Miras-Moreno, B., Rocchetti, G., Lucini, L., Hauser, M. T., & Ranieri, A. (2020). Beyond the visible and below the peel: How UV-B radiation influences the phenolic profile in the pulp of peach fruit. A biochemical and molecular study. Frontiers in Plant Science, 11, 579063.

Santin, M., Lucini, L., Castagna, A., Rocchetti, G., Hauser, M. T., & Ranieri, A. (2019). Comparative “phenol-omics” and gene expression analyses in peach (Prunus persica) skin in response to different postharvest UV-B treatments. Plant physiology and biochemistry, 135, 511-519.

Reviewer 4 Report

Research questions are well defined and within the aims and the scope of the journal. Material is mainly accordingly defined. Methods are suitable, properly described and used in a way that is possible to replicate experiments and analyses. The investigation is performed to good technical standards. It is no ethical problem involved. Conclusions are well stated and based on the results. Discussion is sound and relevant.

Suggestions:

Line 34. Delete »Plants and light constitute inseparable companions« or improve the style.

Line 424. »uniform in colour«. Please define it. Green? Yellow?

Section 4.1. Define in more details the time of irradiance.

Line 440. How deep were taken the samples of the peel?

Line 443. Which are »the relevant guidelines«? Please give details.

Line 424. What was the elevation of the farm Azienda? In the discussion should be discussed possible impact of previous UV irraditation of plants/fruits.

Author Response

Reviewer 4

Research questions are well defined and within the aims and the scope of the journal. Material is mainly accordingly defined. Methods are suitable, properly described and used in a way that is possible to replicate experiments and analyses. The investigation is performed to good technical standards. It is no ethical problem involved. Conclusions are well stated and based on the results. Discussion is sound and relevant.

Response from the authors: We thank the Reviewer for his/her positive evaluation of our manuscript.

Suggestions:

Line 34. Delete »Plants and light constitute inseparable companions« or improve the style.

Response from the authors: We have deleted the sentence accordingly.

Line 424. »uniform in colour«. Please define it. Green? Yellow?

Response from the authors: We have specified this, adding the sentence in brackets “mostly yellow with few orange spots”

Section 4.1. Define in more details the time of irradiance.

Response from the authors: We do not understand which details about the time of irradiance the Reviewer asks for. We have provided the duration of the UV-B exposure (60 min) and the corresponding UV-B dose given (38.53 kJ m−2, consisting of 8.33 kJ m−2 UV-B + 30.20 kJ m−2 white light) as described in material and methods section.

Line 440. How deep were taken the samples of the peel?

Response from the authors: The whole peel tissue was sampled, being careful to remove any pulp traces. Thus, the thickness of the samples corresponds exactly to the thickness of just the peel.

Line 443. Which are »the relevant guidelines«? Please give details.

Response from the authors: for “relevant guidelines” we intended all the activities related to the collection, manipulation, and use of plants that have been carried out respecting the rights and needs of individuals involved in the research and the adherence to standard safety protocols. We understand that the sentence could be misleading, for this reason we decided to remove it from the manuscript.

Line 424. What was the elevation of the farm Azienda? In the discussion should be discussed possible impact of previous UV irraditation of plants/fruits.

Response from the authors: according to the Reviewer’s comment, we have added the requested information within the Materials and Methods section. Since the farm is located a very low altitude, we do not believe that the ambient UV-B conditions in pre-harvest might have impacted the molecular responses of the post-harvest UV-B treatment. In addition, all the peaches (both controls and UV-B-treated) were subjected to the same solar UV irradiance, therefore the differences in the response are very likely due only to the post-harvest UV-B exposure.

Round 2

Reviewer 1 Report

The manuscript was totally improved and it is now suitable for publication